# Structural Insights and Calcium-Switching Mechanism of *Fasciola hepatica* Calcium-Binding Protein FhCaBP4

**DOI:** 10.3390/ijms26157584

**Published:** 2025-08-05

**Authors:** Byeongmin Shin, Seonha Park, Ingyo Park, Hongchul Shin, Kyuhyeon Bang, Sulhee Kim, Kwang Yeon Hwang

**Affiliations:** 1Department of Biotechnology, College of Life Sciences and Biotechnology, Korea University, Seoul 02841, Republic of Korea; kingboom2@naver.com (B.S.); psh3810@korea.ac.kr (S.P.); goodwill1218@naver.com (I.P.); saekomi5@korea.ac.kr (H.S.); qptmh360@naver.com (K.B.); 2Korea BioDefense Research Institute, Korea University, Seoul 02841, Republic of Korea

**Keywords:** *Fasciola hepatica*, FhCaBP4, EF-hand, DLC-like domain, calcium, homodimer

## Abstract

*Fasciola hepatica* remains a global health and economic concern, and treatment still relies heavily on triclabendazole. At the parasite–host interface, *F. hepatica* calcium-binding proteins (FhCaBPs) have a unique EF-hand/DLC-like domain fusion found only in trematodes. This makes it a parasite-specific target for small compounds and vaccinations. To enable novel therapeutic strategies, we report the first elevated-resolution structure of a full-length FhCaBP4. The apo structure was determined at 1.93 Å resolution, revealing a homodimer architecture that integrates an N-terminal, calmodulin-like, EF-hand pair with a C-terminal dynein light chain (DLC)-like domain. Structure-guided in silico mutagenesis identified a flexible, 16-residue β4–β5 loop (LTGSYWMKFSHEPFMS) with an FSHEPF core that demonstrates greater energetic variability than its FhCaBP2 counterpart, likely explaining the distinct ligand-binding profiles of these paralogs. Molecular dynamics simulations and AlphaFold3 modeling suggest that EF-hand 2 acts as the primary calcium-binding site, with calcium coordination inducing partial rigidification and modest expansion of the protein structure. Microscale thermophoresis confirmed calcium as the major ligand, while calmodulin antagonists bound with lower affinity and praziquantel demonstrated no interaction. Thermal shift assays revealed calcium-dependent stabilization and a merger of biphasic unfolding transitions. These results suggest that FhCaBP4 functions as a calcium-responsive signaling hub, with an allosterically coupled EF-hand–DLC interface that could serve as a structurally tractable platform for drug targeting in trematodes.

## 1. Introduction

*Fasciola hepatica* is a globally distributed food-borne trematode that damages the liver and bile ducts of humans and livestock [1]. Although it is officially categorized as a neglected tropical disease, fascioliasis leads to serious chronic illness in endemic areas and costs the livestock industry billions every year [2]. Treatment still relies predominantly on triclabendazole (TCBZ). However, field reports and population studies show increasing TCBZ treatment failures [3,4].

Recent genetic studies demonstrate that resistance can arise via dominant inheritance at a major locus and through multiple, non-parallel evolutionary routes, which makes depending on a single drug a risky strategy [4,5]. Given the risk of resistance, it is crucial to find parasite-specific drug targets that the organism needs for survival and immune evasion [6]. Proteomic work on *F. hepatica* secretions and extracellular vesicles (EVs) shows that several proteins are connected to calcium signaling and host interaction, especially from the parasite’s tegument and gut. These findings suggest that calcium-sensitive proteins could help the parasite modulate the host immune system [7,8,9,10]. Still, high-resolution structural data for full-length trematode calcium-binding proteins are scarce, which limits mechanistic understanding or design structure-based treatments [11,12].

A unique family of proteins called calcium-binding protein (CaBPs) or tegumental allergen-like (TALs) proteins are abundant in trematodes. These proteins have a dynein light chain (DLC)-like domain at the C-terminal and a calmodulin-like domain with an EF-hand motif at the N-terminal [11]. Biochemical work on *F. hepatica* shows that each paralog (FhCaBP1-4) behaves differently in terms of ion and small-molecule binding, oligomer formation, and surface hydrophobicity. These differences point to distinct signaling roles for each one [13,14,15,16]. So far, only the isolated DLC-like domain of FhCaBP2 has been structurally characterized and full-length structures for *Fasciola* CaBPs have been lacking, which are necessary to understand how the EF-hand and DLC-like domains work together and rationalize calcium switching and ligand selectivity [12].

Among the *F. hepatica* paralogs, FhCaBP4 is notable because unlike other proteins in this group, it exhibits calcium-dependent dimerization and increases hydrophobic surface exposure upon calcium ion binding [16]. This behavior may demonstrate that it uses calcium as a switch for conformational change. Studies on a related protein in *Fasciola gigantica* show that CaBP4 orthologs can influence monocyte nitric oxide and cytokine levels, supporting a role in immune modulation and hints at possible therapeutic relevance [17].

Here, we report the high-resolution structure of full-length FhCaBP4 in its apo form to gain a better understanding of the trematode-specific calcium switch and its potential for medication or vaccine development at the molecular level. We also analyze how its EF-hand and DLC-like domains interact, especially in the context of calcium-induced switching. Ligand-binding and thermal shift assays were used to test specificity for calcium relative to known drugs including calmodulin antagonists.

## 2. Results

### 2.1. Overall Structure of the Apo FhCaBP4

To investigate the structural basis underlying calcium-dependent activity of FhCaBP4, we attempted to crystallize the full-length protein (residues 1–191) in its calcium-bound form. We successfully determined the crystal structure of the apo form at 1.93 Å resolution. The asymmetric unit contained two FhCaBP4 molecules forming a homodimer with a two-fold axis (Figure 1A,B).

The crystal belongs to the orthorhombic space group P2_1_2_1_2_1_ with unit cell dimensions a = 62.25 Å, b = 71.59 Å, and c = 93.80 Å, as well as angles α = β = γ = 90°. The structure demonstrated excellent refinement statistics, with acceptable deviations in bond lengths, angles, and geometric parameters (Table 1).

Each FhCaBP4 monomer consists of an N-terminal calmodulin-like domain (residues 4–90) connected via a 10-residue flexible hinge to a C-terminal dynein light chain (DLC)-like domain (residues 101–190). The calmodulin-like domain comprises five α-helices. The α1 to α4 helices form classical EF-hand motifs, where two helix–loop–helix structures are connected by two short antiparallel β-sheets (β1 and β2) that flank the Ca^2+^-binding loops. The DLC-like domain adopts a compact β-sandwich fold composed of antiparallel β-strands (β3, β5, β6, β4) capped by an α6/α7 helical hairpin (Figure 1A,C).Sequence conservation analysis revealed that the EF-hand motifs are genetically conserved, whereas the flexible linker region is not, suggesting that this region may confer species-specific flexibility and functional diversity (Appendix A).

The dimeric structure of FhCaBP4 exhibits similar dimerization interfaces to previously reported DLC-domain dimers [12,18,19]. Five residues (W163 to S167) of each protomer in the extended loop interact with five residues (Q’154 to V’157) of the neighboring protomer in the β4-strand through a pair of hydrogen bonds. Additionally, side chains from these strands contribute to hydrophobic interactions (Figure 2A,B). 

The DLC-like domain of FhCaBP4 shares elevated structural similarity with other known DLC domains. Notable examples include LC8 from Drosophila melanogaster (3BRI, RMSD 1.58 Å over 80 Cα atoms), TgDLC8a from Toxoplasma gondii (3RJS, RMSD 1.49 Å over 80 Cα atoms), dynein light chain from Saccharomyces cerevisiae (4DS1, RMSD 1.39 Å over 80 Cα atoms), and the DLC-like domain of CsTAL3 (5X2D, RMSD 2.31 Å over 88 Cα atoms) in PyMOL [20]. Superposition with the FhCaBP2 DLC-like domain model (5XF0) yielded an RMSD of 1.45 Å over 88 Cα atoms in PyMOL and 1.0 Å with a Z-score of 16.1 for 87 Cα atoms in DALI [21], highlighting strong conservation within the FhCaBP family (Figure 2C).

### 2.2. Structural Comparison of FhCaBP4 with Its Homolog

As previously noted, FhCaBP2 and FhCaBP4 share a highly similar amino acid sequence. However, a key deviation was found in the flexible loop between β4 and β5 (residues Phe166 to Phe171), which displayed significant conformational variability despite the conserved sequence (Figure 1A and Figure 3A). In FhCaBP4, 12 residues interact within and around this flexible loop. Specifically, the side chain of Glu12 forms a hydrogen bond with the primary chain of Phe171, and Thr107 forms one with Ser173. Glu169 also forms a hydrogen bond with Arg187, a critical residue that itself forms a strong salt bridge with Glu78. Additional backbone interactions include Pro170 with Met172 and Glu169 with Met172. Furthermore, Trp186 engages in a T-shaped π–π interaction with Phe166 (Figure 3B). In contrast, FhCaBP2 exhibits only four intramolecular contacts in this region. These include hydrogen bonding between the side chain of Ser165 and the primary chain of Glu167, interactions of the Ser171 primary chain with Arg185 and Thr105, and a π–π interaction between Phe169 and Phe164 (Figure 3C). The elevated number of interactions in FhCaBP4, particularly those involving Glu12 and Glu78 located in helices α1 and α5, respectively, indicates that the calmodulin-like domain contributes to stabilizing the flexible loop within the DLC-like domain.

To further investigate this divergence, site-saturation mutagenesis was performed on the 16-residue loop (LTGSYWMKFSHEPFMS) in FhCaBP4 and the homologous sequence (LTGSYWMHFSHEPFLS) in FhCaBP2. Each residue was substituted with the remaining 19 amino acids, and ΔΔGbind values were estimated by using MMGBSA [22]. Violin plots demonstrated that FhCaBP4 and FhCaBP2 exhibited distinct ΔΔGbind distribution patterns, with FhCaBP4 displaying broader distributions at most positions. The FSHEPF segment (positions 9–14), in particular, demonstrated significantly higher variability in FhCaBP4 (*p* < 0.001), with greater mean ΔΔGbind and wider spread (Figure 4). Interestingly, position 14 (Phe) in FhCaBP2 exhibited higher ΔΔGbind values and wider variance than in FhCaBP4. This difference is likely attributable to its involvement in strong π–π interactions with an adjacent phenylalanine, a structural feature that FhCaBP4 does not possess.

As described above, structural analysis reveals that the 16-residue loop between β4 and β5 engages in distinct interactions in the two homologs. In FhCaBP4, Phe166 forms a π–π stack and Glu169 forms a hydrogen bond, while in FhCaBP2, Phe164 forms a π–π stack with Phe169. These features reinforce the energetic disparities observed in mutagenesis data. Thus, FSHEPF is a critical determinant of loop flexibility and binding affinity. This structural plasticity could influence not only calcium binding but also interactions with small molecules such as calmodulin antagonists and praziquantel (PZQ). This flexibility might account for the diversity of ligand-binding profiles seen in the FhCaBP family and related proteins in the TAL in *Schistosoma mansoni* (SmTAL) family [12,14,15,23,24].

### 2.3. Calcium-Induced Structural and Dynamical Changes

To examine potential calcium-bound conformations, the full-length FhCaBP4 structure with Ca^2+^ coordination was predicted by using the AlphaFold 3 web server (Google DeepMind Technologies Ltd., London, UK) [25]. The model exhibited high elevated global confidence, with pLDDT scores exceeding 90 (residues 1–191), an ipTM of 0.94, a pTM of 0.81, and favorable PAE values (Appendix A). AlphaFold 3 predicts that the protein possesses a calmodulin-like domain containing two EF-hand motifs at the N-terminus (residues 1–90) and a DLC-like domain at the C-terminus (residues 101–191). These two domains are predicted to fold independently and are connected by a flexible linker. An electrostatic analysis of the predicted calcium-bound model demonstrated that the surfaces of EF-hand motifs, especially helices α2 and α3, are negatively charged, suggesting their role in calcium binding (Figure 5A).

AlphaFold3 predicts two EF-hand motifs coordinating calcium ions. EF-hand 1 binds Ca^2+^ via D22, N24, D26, I28, D30, and E33, while EF-hand 2 coordinates Ca^2+^ via D56, D58, N60, K62, T64, and E67 (positions X, Y, Z, −X, −Y, and −Z, respectively), forming a pentagonal bipyramidal geometry (Figure 5B). D56, D58, N60, and E67 coordinate the ion via side chain oxygens, while K62 binds through its primary-chain carbonyl. Although T64 (−Y position) is slightly distant, previous structural surveys have shown that EF-hand loops tolerate such deviations through backbone shifts or water coordination, maintaining functional geometry [26,27,28]. Among the two predicted EF-hand motifs, EF-hand 2 was selected as the functionally relevant calcium-binding site. Structural alignment with the Reps1 EH domain (PDB: 1FI6) and similar EH family proteins (TPLATE EH1.2, EHD/Eps15) demonstrates that calcium binding occurs exclusively in EF-hand loop 2 [29,30,31]. Mutation studies in trematodes support this view. A D58A substitution in loop 2 of FhCaBP2 abolishes calcium binding, while mutations in loop 1 have no functional impact [13]. Sequence analyses confirm that FhCaBP4 loop 1 lacks several canonical calcium-coordinating residues [26,32]. Similarly, calcium binding in other EF-hand proteins such as polycystin-2 and parvalbumin is restricted to the second EF-hand [33,34]. These findings support a model in which FhCaBP4 functions as a single-site calcium sensor, with EF-hand 2 serving as the principal binding site for calcium.

Superposition of the calcium-bound AlphaFold3 model and the apo crystal structure revealed domain-specific conformational rearrangements. The calmodulin-like domain exhibited an RMSD of 3.5 Å (79 Cα atoms, DALI Z-score 7.6), whereas the DLC-like domain remained structurally stable (RMSD 0.5 Å, 90 Cα atoms, DALI Z-score 20.0). Notably, helices α2 and α3 moved outward, exposing hydrophobic patches (Appendix A) and forming a canonical coordination cage with residues D56, D58, N60, K62, T64, and E67 (Figure 5C,D). These findings indicate that calcium binding alters the spatial configuration of the calmodulin-like domain without significantly affecting the DLC-like domain.

To assess how calcium binding influences structural dynamics, molecular dynamics (MD) simulations by using GROMACS were performed for both apo and Ca^2+^-bound forms at 300 K and 1 atm for 30 ns [35]. RMSD plots confirmed that both forms remained stable after 10 ns of simulation, with average RMSDs of 0.3 nm (apo) and 0.2 nm (Ca^2+^-bound) (Figure 6A). The root mean square fluctuation (RMSF) analysis revealed that, although overall fluctuation profiles were similar, key differences appeared in loop regions. In the apo state, increased fluctuations were observed in five regions: the loops between helices α1–α2 (residues 22–30), helices α3–α4 (residues 52–60), the linker between helix α5 and β-strand β3 (residues 91–99), the α6–α7 loop (residues 126–134), and the β4–β5 loop (residues 158–171) (Figure 6B). In contrast, the Ca^2+^-bound state demonstrated pronounced fluctuations in the region spanning the loop between helices α2 and α3 (residues 35–48) as well as parts of the helices themselves. Compared to the apo form, moderate reductions in fluctuation were observed in the linker between helix α5 and β-strand β3 (residues 91–99) and in the α6–α7 loop (residues 126–134). These findings suggest that calcium binds stably at EF-hand 2, and the resulting movement of the N-terminal calmodulin-like domain may influence the dynamics of the C-terminal DLC-like domain. This interplay may, in turn, modulate binding interactions with other drugs or small molecules. The compactness of the apo and Ca^2+^-bound states were evaluated by calculating the radius of gyration (Rg). Throughout the 30 ns MD simulation, Rg values for both states ranged from 1.8 nm to 1.95 nm. The Ca^2+^-bound form demonstrated an approximately 0.05 nm higher Rg value than the apo form. This observation is consistent with the behavior of many EF-hand proteins, which undergo a transition from a closed to an open conformation upon calcium binding, thereby exposing hydrophobic surfaces for target recognition and causing a slight increase in Rg.

The correlation between the structural and dynamic changes observed in helices α2 and α3 supports a model in which calcium-induced conformational rearrangements drive functional switching. This mechanism is comparable to those previously reported in calmodulin and SmTAL family proteins [23,36,37]. These findings indicate that calcium binding induces a major conformational shift in FhCaBP4 that likely facilitates interactions within the parasite tegument and contributes to calcium-regulated signaling pathways essential for parasite survival.

### 2.4. Drug-Binding and Thermal Stability Analysis

To characterize the ligand-binding profile of FhCaBP4, microscale thermophoresis (MST) assays were conducted with Ca^2+^, calmodulin antagonists (W7, CPZ, TFP), and the antiparasitic agent PZQ. FhCaBP4 exhibited the highest affinity for Ca^2+^ (Kd = 43.1 ± 21.1 µM), followed by CPZ (288.9 ± 99.6 µM), W7 (470.9 ± 157.3 µM), and TFP (699.3 ± 341.2 µM). PZQ demonstrated negligible binding (Kd > 2000 µM) (Figure 7A–E). These findings confirm Ca^2+^ as the primary physiological ligand of FhCaBP4. The moderate binding of calmodulin antagonists indicates weak mimicry of Ca^2+^-binding interactions, while the lack of PZQ interaction suggests that FhCaBP4 is not a direct target of this drug, setting it apart from other calcium-binding proteins involved in anthelmintic mechanisms.

To evaluate how ligand binding affects thermal stability, we performed thermal shift analysis by using the Tycho NT.6 instrument, which tracks temperature-dependent intrinsic fluorescence at 330 and 350 nm. Inflection temperatures (Ti) were derived from the 350/330 nm fluorescence ratio (Appendix A, Table 2). Wild-type FhCaBP4 displayed two inflection points at 49.5 ± 0.5 °C and 68.0 ± 0.4 °C, reflecting the independent unfolding of distinct domains. Ligand addition affected thermal stability to varying extents. PZQ, CPZ, and TFP slightly decreased the first Ti to 49.1 ± 1.8 °C, 47.8 ± 1.8 °C, and 46.8 ± 1.2 °C, respectively. In contrast, the second Ti shifted upward to 69.5 ± 1.3 °C, 69.3 ± 0.7 °C, and 68.5 ± 1.8 °C, respectively. W7 did not produce a detectable inflection point, possibly due to poor solubility or specific binding that interferes with thermal unfolding, potentially through inhibition of proteolysis and stabilization of the overall protein [16,38]. In contrast, Ca^2+^ treatment yielded a single, pronounced Ti at 76.2 ± 1.1 °C, indicating marked stabilization of FhCaBP4. Given that FhCaBP4 comprises distinct calmodulin-like and DLC-like domains, the observation of two Ti values in the apo state likely reflects independent domain unfolding. The shift to a single Ti upon calcium binding suggests that Ca^2+^ not only stabilizes the calmodulin-like domain but also imposes structural reinforcement across the DLC-like domain through allosteric effects.

According to the MST results, calmodulin antagonists (W7, CPZ, TFP) and PZQ bind weakly or not at all, consistent with the minor or negligible shifts observed in thermal stability. Conversely, calcium demonstrated strong binding and significantly increased protein stability. These data collectively support the notion that calcium plays a central role in stabilizing FhCaBP4 by binding to its calmodulin-like domain and inducing allosteric stabilization across the entire protein. Thus, while some calmodulin antagonists weakly associate with FhCaBP4 and slightly modulate thermal behavior, only calcium substantially enhances thermal resilience. The lack of PZQ interaction suggests selective ligand recognition among calcium-binding proteins in parasitic systems. These insights could guide structure-based drug design strategies aimed at selectively targeting calcium-binding domains unique to trematode parasites.

## 3. Discussion

We determined the first full-length crystal structure of a CaBP4 from *F. hepatica* (Figure 1 and Figure 2). The structure shows a conserved EF-hand calcium-binding motif and a DLC-like domain involved in dimerization bound together in a polypeptide. The full-length structure of FhCaBP2 was examined, although only the C-terminal electron density was detected [12]. It was proposed that FhCaBP2 can be readily cleaved into two domains by protease for cell extraction. Comparable findings were seen for the tegumental-allergen-like (TAL) proteins derived from *Clonorchis sinensis* (CsTALs) [12,14,19]. FhCaBP4 belongs to a trematode-specific family of calcium-binding proteins that function as calcium-sensitive signaling platforms. Comparative studies in other trematodes, such as the TALs in *Schistosoma mansoni* (SmTALs) and FhCaBPs, have shown similar EF-hand/DLC-like domain combinations, with paralog-specific differences in calcium responsiveness and drug interaction profiles [13,14,15,23].

In the calcium-binding model of AlphaFold3, EF-hand 2 adopts canonical coordination and superposition with the apo structure, showing an outward motion of helices including EF-hand 2 as well as increased hydrophobic exposure (Figure 5B). Also, MD analyses show that calcium binding slightly increases Rg and redistributes loop dynamics. These data suggest that EF-hand 2 acts as the primary calcium-binding site and is compatible with a conformational shift brought on by calcium. The flexible β4–β5 loop in FhCaBP4, which differs from its FhCaBP2 paralog, bridges EF-hand helices to the loop of the DLC-like domain and an added π–π stack stabilizes the loop (Figure 4 and Figure 5). Consistently, site-saturation mutagenesis across the 16-residue loop shows broader ΔΔG distributions in FhCaBP4, with the FSHEPF core displaying significantly greater energetic variability than in FhCaBP2, supporting a paralog-specific plasticity module that likely adjusts ligand and partner interactions (Figure 4).

Our ligand-binding assays demonstrate that calcium binds well and clearly affects the protein, but other molecules like the calmodulin antagonists bind weakly at best (Figure 7). Recently, PZQ primarily targets the TRPM ion channel of a parasite, and the ortholog in Fasciola exhibits sequence variations that account for PZQ insensitivity [39,40]. Figure 7 illustrates that PZQ exhibited minimal binding (Kd > 2000 µM). FhCaBP4 demonstrated the greatest affinity for Ca^2+^ (Kd = 43.1 ± 21.1 µM), succeeded by CPZ (288.9 ± 99.6 µM), W7 (470.9 ± 157.3 µM), and TFP (699.3 ± 341.2 µM). The observed inflection temperature demonstrates that calcium binding possesses strong affinity and significantly improves protein stability (Table 2). Consequently, these results indicate an alternative function and facilitate the targeting of FhCaBP4 as a component of a parasite-selective approach, utilizing either small-molecule inhibitors or vaccine candidates. Additional comparative analysis of FhCaBP paralogs and other trematode calcium-binding proteins, such as SmTALs or CsTALs, may elucidate if the calcium-mediated alteration serves as a prevalent regulatory mechanism within this family, potentially revealing novel functional or therapeutic insights.

## 4. Materials and Methods

### 4.1. Cloning and Recombinant Expression of FhCaBP4

The gene encoding *Fasciola hepatica* calcium-binding protein 4 (FhCaBP4; UniProt ID: I6U578) was amplified by polymerase chain reaction (PCR) by using primers containing restriction enzyme sites for cloning. The PCR product was digested with appropriate restriction enzymes and ligated into the pET-21a (+) vector (Novagen, Sigma-Aldrich, Burlington, MA, USA), thereby introducing a C-terminal 6×His-tag (LEHHHHHH) to facilitate affinity purification. Sequence integrity of the final construct was confirmed by Sanger sequencing (Bionics, Seoul, Republic of Korea) before transformation. Recombinant plasmids were transformed into Escherichia coli BL21 (DE3) cells. A single transformant colony was inoculated into LB medium supplemented with 50 μg/mL ampicillin and cultivated at 37 °C with shaking (180 rpm) until the optical density at 600 nm reached approximately 0.6. Protein expression was induced by the addition of 1 mM isopropyl β-D-1-thiogalactopyranoside (IPTG), followed by incubation at 18 °C for 18 h. Cells were harvested by centrifugation at 3500× *g* for 15 min at 4 °C and stored at −20 °C until purification.

### 4.2. Protein Purification and Crystallization

All purification procedures were conducted at 4 °C unless otherwise specified. Bacterial pellets derived from 2 L cultures were resuspended in 60 mL of lysis buffer containing 20 mM Tris–HCl (pH 8.5), 150 mM NaCl, and 5 mM β-mercaptoethanol (βME). Cells were lysed by ultrasonication on ice by using 30 cycles of 1 s on and 4 s off. The lysate was clarified by centrifugation at 13,000 rpm for 60 min, and the resulting supernatant was filtered through a 0.45 µm polyvinylidene difluoride (PVDF) syringe filter (Merck Millipore, Seoul, Republic of Korea). The filtered lysate was applied to a 5 mL HisTrap HP nickel affinity column (GE Healthcare, Seoul, Republic of Korea) pre-equilibrated with binding buffer (20 mM Tris–HCl, pH 8.5; 150 mM NaCl; 5 mM βME). Bound proteins were eluted by using a linear gradient of imidazole in elution buffer composed of 20 mM Tris–HCl (pH 8.5), 150 mM NaCl, 5 mM βME, and 500 mM imidazole. Elution fractions were analyzed by SDS–PAGE to confirm the presence of FhCaBP4 at the 23 kDa. The eluted protein was further purified by size-exclusion chromatography (SEC) by using a Superdex 75 Increase 10/300 GL column (GE Healthcare, Seoul, Republic of Korea) equilibrated in SEC buffer consisting of 20 mM Tris–HCl (pH 8.5), 150 mM NaCl, and 2 mM dithiothreitol (DTT). Peak fractions corresponding to the monomeric form were pooled and concentrated to 62 mg/mL by using a 10 kDa molecular weight cutoff Amicon Ultra centrifugal filter unit (Merck Millipore, Seoul, Republic of Korea).

Initial crystallization screening was performed by using the sitting-drop vapor diffusion method at 20 °C in MRC 2-well plates (Hampton Research, Seoul, Republic of Korea). Protein at 10 mg/mL was mixed in a 1:1 ratio with commercial reservoir solutions from sparse-matrix screens (e.g., Crystal Screen and PEG/Ion, Hampton Research). Crystals of FhCaBP4 formed under multiple conditions containing ammonium sulfate as the precipitant. Optimization of initial hits yielded the best diffracting crystals by using 2 M ammonium sulfate and 0.1 M sodium citrate/citric acid (pH 5.5) as the reservoir solution. Hanging-drop vapor diffusion was subsequently used to grow larger single crystals suitable for diffraction analysis by mixing 1 μL protein solution with 1 μL optimized reservoir solution and equilibrating against 500 μL of the same reservoir at 20 °C.

### 4.3. X-Ray Data Collection

Before data collection, crystals were cryoprotected by immersion in reservoir solution supplemented with 25% (*v/v*) glycerol for 5–10 s and then flash-cooled in liquid nitrogen. X-ray diffraction data were collected at 100 K on beamline BL-11C at the Pohang Accelerator Laboratory (PAL, Pohang, Republic of Korea) and BL-44XU at Spring-8 (Sayo, Japan) by using a monochromatic wavelength of 1.000 Å. A complete data set was collected on the PILATUS3 × 6 M detector by using 1° oscillation and 1 s exposure per image over a total of 360°. Data were indexed, integrated, and scaled by using the HKL-2000 software suite [41].

### 4.4. Structure Solution and Refinement

The crystal structure of FhCaBP4 was solved by molecular replacement (MR) by using the Phaser module in the PHENIX software (version 1.21) [42]. Due to the absence of a closely homologous full-length structure in the Protein Data Bank (PDB), a predicted structural model of FhCaBP4 was generated by using the AlphaFold3 server (https://alphafold.ebi.ac.uk) [25]. The highest-ranked model was manually edited by truncating flexible loops and used as the MR search model. Phaser identified a clear MR solution in the orthorhombic space group P2_1_2_1_2_1_, with two FhCaBP4 molecules per asymmetric unit forming a dimer. Iterative cycles of manual model building and refinement were performed by using COOT (0.9.8.95) [43] and phenix.refine (version 1.21). Refinement strategies included positional refinement, isotropic B-factor refinement, and translation–libration–screw (TLS) parameterization. Non-crystallographic symmetry (NCS) restraints were applied during early refinement stages and gradually relaxed. Calcium ions were not included in the final model due to the absence of strong peaks in the Fo–Fc electron density maps at EF-hand sites, consistent with the protein being in apo state. Water molecules were automatically placed by using PHENIX and subsequently curated manually. The final refined model yielded R_work and R_free values of 20.9% and 25.1%, respectively. Structural validation with MolProbity [44] indicated that 98.6% of residues were located within favored Ramachandran regions, with no outliers, and root mean square deviations (RMSDs) of 0.010 Å for bond lengths and 1.25° for bond angles.

### 4.5. Structural Analysis

Structure-based sequence alignments were generated by using Clustal Omega 1.2.4 [45] and visualized with ESPript 3.0 [46]. Structural similarity metrics, including root mean square deviation (RMSD) and Z-score, were obtained by using the DALI server [21]. Residue conservation scores were calculated based on Shannon entropy to assess sequence variability [47]. Site-saturation mutagenesis was performed in silico for the 16-residue loop by using the BeAtMuSiC server (v1.0) [48]. Each residue was systematically substituted with all 19 other amino acids. The change in binding free energy (ΔΔGbind) relative to the wild type was calculated by using the MM/GBSA protocol [22] implemented in BeAtMuSiC, with the AMBER ff14SB force field, the igb = 2 generalized Born solvation model, 0.15 M salt concentration, and default parameters for atomic radii and dielectric constants.

### 4.6. Statistical Analysis

Comparative statistical analyses of ΔΔGbind distributions at each loop position were conducted by using two-sided Wilcoxon rank-sum tests [49]. Significance thresholds were designated as follows: *** *p* < 0.001, ** *p* < 0.01, * *p* < 0.05, and ns (not significant) for *p* ≥ 0.05. All tests were performed without assuming equal variances.

### 4.7. Molecular Modeling

To generate a model of the protein with a calcium ion, the full-length FhCaBP4 sequence (UniProt: I6U578) was submitted to the AlphaFold 3 web server (Google DeepMind Technologies Ltd., London, UK) by using default diffusion-based parameters that explicitly sample calcium coordination states [25]. Five ranked models were returned, and we chose the highest-rank calcium-bound FhCaBP4 model. Coordinates for the calcium-bound model were used in subsequent comparison and structure analyses.

### 4.8. MD Simulation

All-atom molecular dynamics simulations were performed by using GROMACS 2021.4 [35]. The protein was modeled with the CHARMM36m force field and solvated in a dodecahedral TIP3P water box, extending 10 Å beyond the protein surface [50]. Na^+^ and Cl^−^ ions were added to achieve a 0.15 M ionic concentration and to neutralize the system. Following steepest descent energy minimization (Fmax ≤ 1000 kJ mol^−1^ nm^−1^), both systems underwent 0.1 ns simulations under NVT and NPT ensembles at 300 K and 1 atm. Production runs were conducted for 30 ns by using a 2 fs time step, with LINCS constraints on hydrogen atoms and Particle Mesh Ewald (PME) electrostatics (10 Å real-space cutoff) [51]. Coordinates were saved every 10 ps. Backbone RMSD, per-residue RMSF, and radius of gyration (Rg) were calculated by using gmx rms, gmx rmsf, and gmx gyrate, respectively, after alignment to the initial structure. Data were visualized with XmGrace 5.1.25 by using .xvg output files.

### 4.9. Microscale Thermophoresis (MST) Assays

The binding affinities of calcium ions and small molecules (W7, CPZ, TFP, PZQ) to FhCaBP4 were quantified by using microscale thermophoresis (MST). Purified FhCaBP4 was labeled by using the Monolith NT™ His-Tag Labeling Kit (RED-tris-NTA 2nd generation; NanoTemper Technologies, South San Francisco, CA, USA), which targets the C-terminal His-tag via tris-NTA dye conjugation. Labeling was carried out according to the manufacturer’s protocol by using 400 nM protein in MST buffer (PBS containing 0.05% Tween-20) and incubated for 30 min at room temperature. Labeling efficiency was confirmed by absorbance spectroscopy.

For calcium-binding assays, a 16-point two-fold serial dilution of CaCl_2_ (ranging from 1 mM to 0.03 μM) was prepared in MST buffer. Labeled FhCaBP4 was diluted to 200 nM in calcium-free MST buffer to ensure the apo form. Equal volumes (10 μL each) of protein and titrant were mixed and incubated briefly before analysis by using a Monolith NT.115 instrument (NanoTemper Technologies, South San Francisco, CA, USA). MST measurements were conducted at 25 °C by using standard capillaries and 40% MST power. Data were analyzed by using MO.Affinity Analysis Software v2.1.3 (NanoTemper Technologies, South San Francisco, CA, USA), and dissociation constants (Kd) were derived by fitting normalized fluorescence (Fnorm) to a 1:1 binding model. The same experimental protocol was applied for drug-binding assays by using W7, CPZ, TFP, and PZQ. Each MST experiment was performed in triplicate to ensure reproducibility.

### 4.10. Thermal Shift Assays

The thermal stability of FhCaBP4 in both apo- and ligand-bound states was evaluated by using the Tycho NT.6 instrument (NanoTemper Technologies, South San Francisco, CA, USA). Protein samples (1 mg/mL in PBS, pH 7.4) were prepared under the following conditions: apo, calcium-bound (1 mM CaCl_2_), and in a complex with small molecules (1 mM each of W7, CPZ, TFP, or PZQ). Samples (10 μL) were loaded into Tycho NT.6 capillaries and subjected to a thermal ramp from 35 °C to 95 °C at a heating rate of 2 °C/min. The intrinsic fluorescence ratio (F_350_/F_330_) was monitored in real time, and inflection temperatures (Ti) were automatically calculated by using Tycho NT.6 Analysis Software (NanoTemper Technologies, South San Francisco, CA, USA).

## 5. Conclusions

First, to investigate novel parasite-specific techniques, we determined the whole structure of FhCaBP4, a calcium-binding protein derived from *F. hepatica*. The structure constitutes a homodimer that connects a calmodulin-like domain, including an EF-hand motif at the N-terminus, to a dynein light chain (DLC)-like domain at the C-terminus. Secondly, we propose that EF-hand 2 serves as the primary calcium-binding site, corroborated by biochemical and biophysical investigations, including AlphaFold3 modeling, molecular dynamics simulations, and microscale thermophoresis experiments. Third, when calcium attaches, it stabilizes the structure and exhibits a more distinct unfolding pattern. Conversely, substances such as calmodulin inhibitors or praziquantel demonstrate minimal or insignificant interaction. Collectively, the findings endorse a calcium-gated switching mechanism that conveys EF-hand movements to the DLC-like domain. This switching interface, absent in mammals and seemingly exclusive to trematodes, may provide a valuable foundation for the development of selective inhibitors or immune-targeting methods to address fascioliasis.

## Figures and Tables

**Figure 1 ijms-26-07584-f001:**
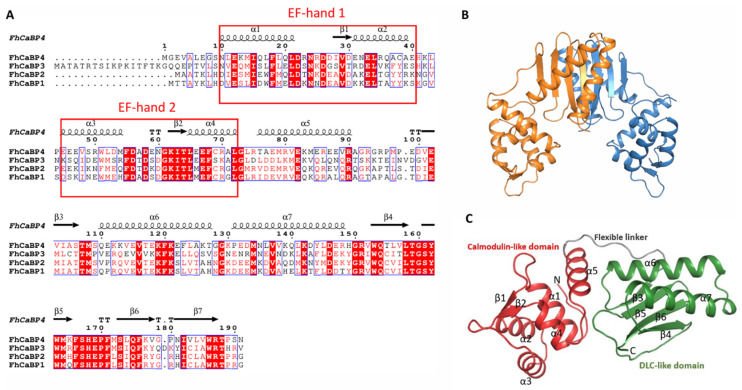
Overall structure of FhCaBP4. (**A**) Multiple-sequence alignment of the four CaBP paralogs (FhCaBP1–4). Positions identical in ≥70% of the sequences are shaded red, while conservative substitutions appear in pink. Secondary-structure elements assigned from the present FhCaBP4 structure are displayed above the alignment (black cylinders, α-helices α1–α6; black arrows, β-strands β1–β7). The two EF-hand calcium-binding loops are boxed in red. (**B**) Dimeric structure of FhCaBP4. The dimeric structure is shown as a cartoon diagram showing with orange and blue, respectively. (**C**) Monomeric structure of FhCaBP4. The calmodulin-like domain is shown in red, the dynein light chain (DLC)-like domain in green, and the intervening Gly-rich linker in gray. Structural elements are labeled sequentially (α1–α7, β1–β6). N- and C-termini are labeled.

**Figure 2 ijms-26-07584-f002:**
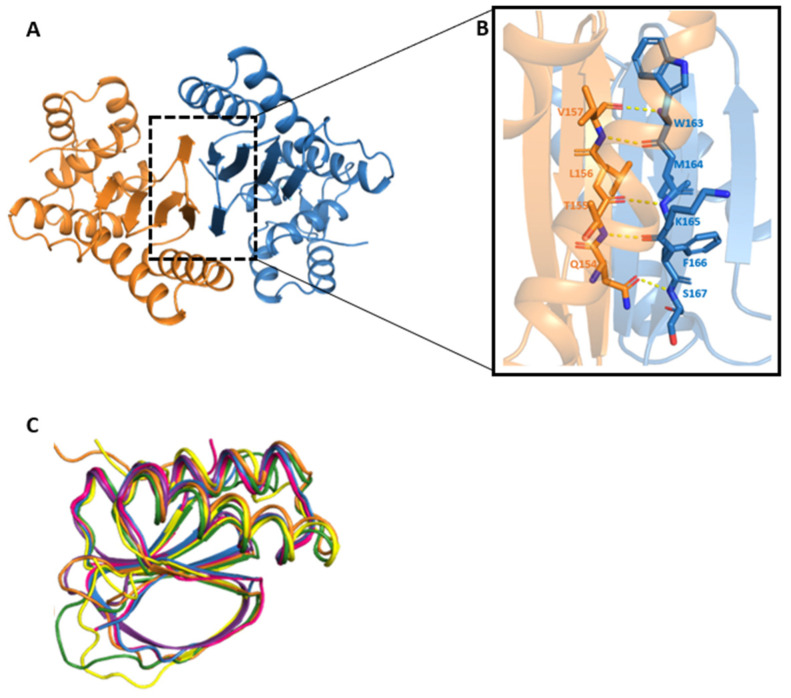
Dimeric structure of the FhCaBP4 and its DLC-like domain (**A**) Dimeric structure of the FhCaBP4. One monomer is colored in orange and the other in sky-blue, represented as a cartoon model. The dimeric interface area is represented by a black dashed rectangle. (**B**) Close-up view of the dimeric interface. Yellow dashed lines represent hydrogen bonds of the primary chains, and the interacting residues are represented as a stick model. (**C**) Superimposition of the DLC-like domains. The DLC-like domain backbone models of *Drosophila melanogaster* (PDB3BRI), *Toxoplasma gondii* (PDB3RJS), *Saccharomyces cerevisiae* (PDB4DS1), *Clonorchis sinensis* (PDB5X2D), *Fasciola hepatica* (PDB5FX0), and FhCaBP4 are shown in blue, pink, purple, yellow, green, and orange, respectively.

**Figure 3 ijms-26-07584-f003:**
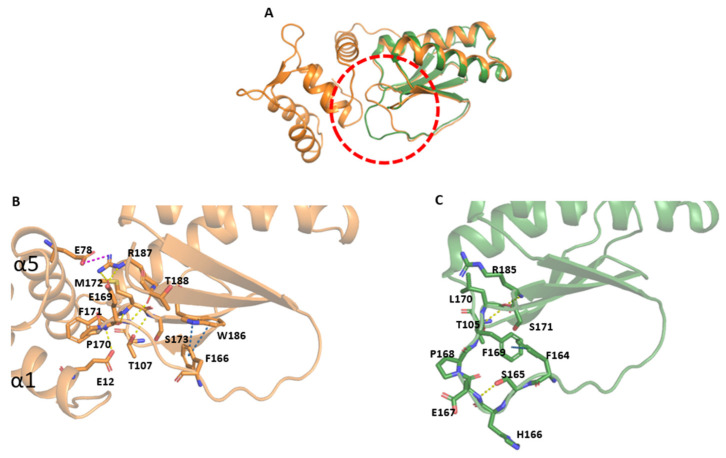
Structural comparison of FhCaBP4 and the DLC-like domain of FhCaBP2. (**A**) Superposition of FhCaBP4 (orange) and FhCaBP2 DLC-like domain (PDB ID: 5FXO; green) displayed in cartoon representation. The flexible loop region is highlighted by a red dashed circle. (**B**) Zoomed-in view of the flexible loop region in the FhCaBP4 structure. Residues involved in hydrogen bonding are connected by yellow dashed lines, salt–bridge interactions by pink dashed lines, and π–π interactions by blue dashed lines. All interacting side chains are shown in stick representation. (**C**) Zoomed-in view of the corresponding loop region in the DLC-like domain of FhCaBP2. Hydrogen bonds are indicated by yellow dashed lines and π–π interactions by blue dashed lines. All interacting side chains are rendered as sticks.

**Figure 4 ijms-26-07584-f004:**
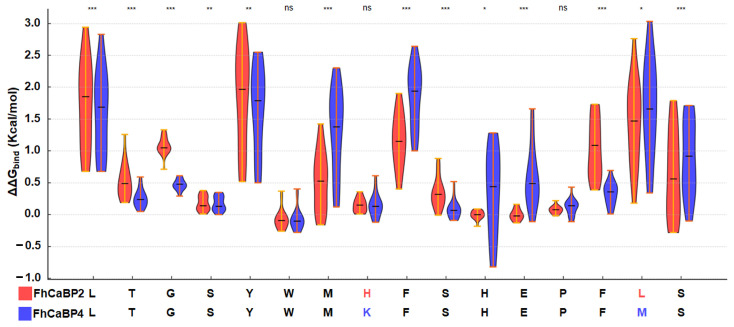
Mutational scanning of the loop in FhCaBP2 (red) and FhCaBP4 (blue). Violin plots show ΔΔGbind distributions when each position in the loop sequence was substituted with the 19 other amino acids in FhCaBP2 and FhCaBP4. Horizontal lines indicate medians. Asterisks indicate significance by two-sided Wilcoxon rank-sum test (*** *p* < 0.001; ** *p* < 0.01; * *p* < 0.05; ns, not significant).

**Figure 5 ijms-26-07584-f005:**
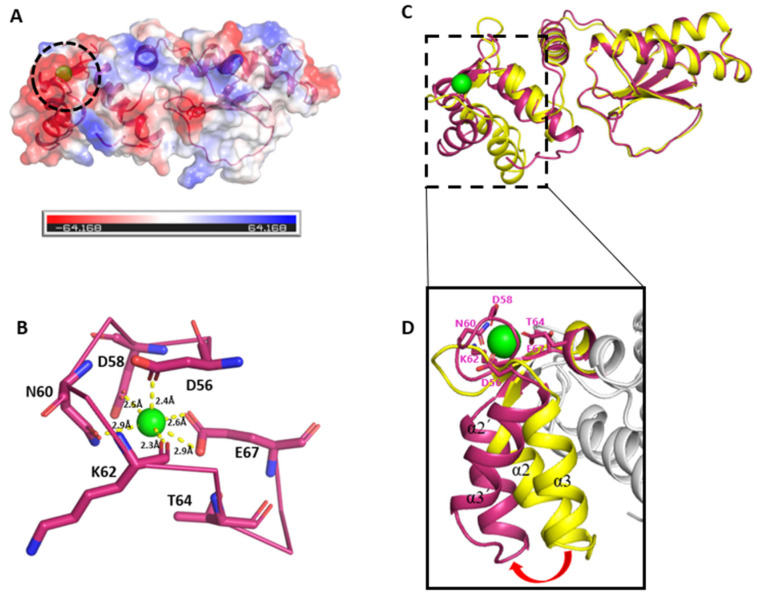
The predicted calcium-bound structure of FhCaBP4 and its conformational changes. (**A**) Electrostatic surfaces of the predicted calcium-bound FhCaBP4. The calcium-binding site is represented by a black dashed circle. (**B**) Clos-up view of the predicted calcium-binding site. The green sphere represents calcium ion. The yellow dashed lines represent the interaction with calcium ions, and residues that interact with calcium are represented as a stick model. (**C**) Superstition of apo-FhCaBP4 (yellow) with an AlphaFold3-predicted calcium-bound model of its calmodulin-like domain (warm pink). The calcium ion at the EF-hand 2 is depicted as a green sphere and the deep view is represented as a black dashed rectangle. (**D**) Close-up view of the moving region. The red arrow highlights an outward rotation of helices α2 and α3 upon calcium binding. The residues that interact with calcium ions are represented as a stick model.

**Figure 6 ijms-26-07584-f006:**
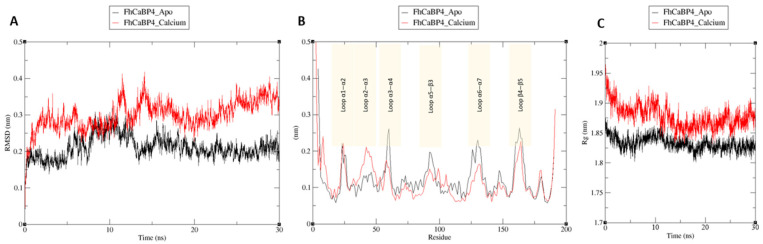
Molecular dynamics analysis of FhCaBP4 apo and calcium-bound form. (**A**) Time-dependent root mean square deviation (RMSD) trajectories for FhCaBP4 apo (black line) and calcium-bound FhCaBP4 (red line) showing backbone stability over 30 ns. (**B**) Residue-wise root mean square fluctuation (RMSF) profiles for the same trajectories. Shaded boxes highlight six loop regions (Loop α1–α2, Loop α2–α3, Loop α3–α4, Loop α5–β3, Loop α6–α7, and Loop β4–β5) where differences in flexibility are most pronounced between apo (black) and calcium-bound (red) forms. (**C**) Radius of gyration (Rg) as a function of simulation time, illustrating that the calcium-bound form (red) maintains a higher average Rg than the apo form (black).

**Figure 7 ijms-26-07584-f007:**
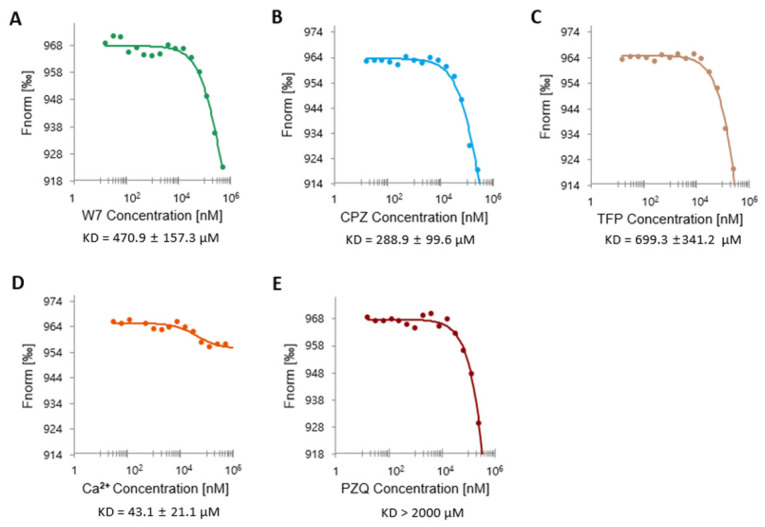
Microscale thermophoresis (MST) analysis of FhCaBP4 binding to chemical ligands. Binding isotherms were generated by plotting normalized fluorescence (Fnorm, %) as a function of increasing ligand concentrations. Each assay was conducted with 100 nM labeled FhCaBP4 and a serial dilution of the corresponding ligand (0.015–500 μM): (**A**) W7, Kd = 470.9 ± 157.3 μM; (**B**) CPZ, Kd = 288.9 ± 99.6 μM; (**C**) TFP, Kd = 699.3 ± 341.2 μM; (**D**) Calcium, Kd = 43.1 ± 21.1 μM; (**E**) PZQ, Kd > 2000 μM. All data represent mean values from at least three independent experiments, each using sixteen capillaries per condition. Dissociation constants (Kd) were calculated by using MO Affinity Analysis software v2.1.3. with a nonlinear quadratic binding model. Values are reported as mean ± SD. μM, micromolar; nM, nanomolar.

**Table 1 ijms-26-07584-t001:** Crystallographic data collection, phase determination, and refinement statistics for apo structure of FhCaBP4.

	Apo
Data collection	
Wavelength (Å)	1.000
Space group	P2_1_2_1_2_1_
Cell dimensions	
a, b, c (Å)	62.25, 71.59, 93.80
α, β, γ (°)	90, 90, 90
Resolution (Å)	42–1.93 (1.999–1.93)
Unique reflections	32,204 (3178)
Completeness	99.95 (99.78)
Redundancy	2.0 (2.0)
I/σ(I)	9.45 (1.25)
R_merge_ (%)	0.04585 (0.4531)
Refinement statistics	
Resolution (Å)	42–1.93
Reflections used in refinement	32,198
R_work_/R_free_ (%)	0.2026/0.2441
R.m.s deviations	
Bond lengths (Å)	0.008
Bond angles (°)	1.04
Ramachandran plot (%)	
Favored	98.65
Allowed	1.35

Statistics for the highest resolution shell are shown in parentheses.

**Table 2 ijms-26-07584-t002:** Thermal unfolding inflection temperatures (Ti) of FhCaBP4 and its ligand-bound complexes at pH 7.4. Reported values are mean ± s.d. from duplicate measurements.

Sample	Ti Value 1 (°C)	Ti Value 2 (°C)
FhCaBP4 WT	49.5 ± 0.5	68.0 ± 0.4
FhCaBP4 + Ca^2+^	76.2 ± 1.1	
FhCaBP4 + PZQ	49.1 ± 1.8	69.5 ± 1.3
FhCaBP4 + CPZ	47.8 ± 1.8	69.3 ± 0.7
FhCaBP4 + TFP	46.8 ± 1.2	68.5 ± 1.8

## Data Availability

The atomic coordinates and structure factors for Apo-FhCaBP4 have been deposited in the Protein Data Bank under accession number PDB 9VOA. All other data supporting the findings of this study are available within the paper and its Appendix A.

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
