# Peer review of "Structural Insights and Calcium-Switching Mechanism of Fasciola hepatica Calcium-Binding Protein FhCaBP4"

_ijms, 2025, doi:10.3390/ijms26157584_

Round 1
Reviewer 1 Report
Comments and Suggestions for Authors
The MS reported the function of FhCaBP4, which is of some interest and contributes to the understanding the pathogenicity of Fasciola hepatica. The followings are my concerns/comments.
There are quite a few formatting and spelling errors. e.g., the species name should be italic, the format of the references is inconsistent in the reference list.
Abstract: there should be a brief introduction about CaBP4 to indcate why study CaBP4 can enable novel therapeutic strategies.
Keywords: FhCaBP4 should be included.
Introduction:
1. The huge paragraph is hard to read. This part should include several paragraphs presented in a logical style to make it more understandable, such as current condition, knowledge gap, what's can we do, the signifcance of this study and what's done in this study.
2. Introduction of FhCaBP should be included, why choose FhCaBP4, but not other members?
3. Figure 1 should be placed in the correct location.
Results and Discussion:
1. Table 1 and 2 is presented in a poor manner, please revised it in a three-line table with necessary note to make it more understandable.
2. As to Figure 5,-7, it will be more interesting to compare FhCaBP4 and FhCaBP2.
3. The authors have explored the structure of FhCaBP4, it will be more interesting to test corresponding small molecule compounds on the FhCaBP4, which likely found potential drugs for treatment.
Conclusion: this part is too long, and should not repeat results, just presented the main fundings and significance of the study in a few sentences.
Supplementary files: all the supplementary files should combined in a separate document.
Author Response
Review1
The MS reported the function of FhCaBP4, which is of some interest and contributes to the understanding the pathogenicity of Fasciola hepatica. The followings are my concerns/comments.
There are quite a few formatting and spelling errors. e.g., the species name should be italic, the format of the references is inconsistent in the reference list.
- We revised them.
Abstract: there should be a brief introduction about CaBP4 to indcate why study CaBP4 can enable novel therapeutic strategies.
- We added the sentences in Abstract. Please see red color in abstract.
“Fasciola hepatica remains a global health and economic concern, and treatment still relies heavily on triclabendazole. At the parasite-host interface, F. hepatica calcium-binding proteins (FhCaBPs) have a unique EF-hand/DLC-like domain fusion found only in trematodes. This makes it a parasite-specific target for small compounds and vaccinations.”
Keywords: FhCaBP4 should be included.
- We added it in Keywords.
Introduction:
1. The huge paragraph is hard to read. This part should include several paragraphs presented in a logical style to make it more understandable, such as current condition, knowledge gap, what's can we do, the signifcance of this study and what's done in this study.
2. Introduction of FhCaBP should be included, why choose FhCaBP4, but not other members?
- We conducted a revision of them. We constructed the sentence for clarity of comprehension.
Figure 1 should be placed in the correct location.
- We changed the location based on the reviewer’s comment.
Results and Discussion:
1. Table 1 and 2 is presented in a poor manner, please revised it in a three-line table with necessary note to make it more understandable.
- Yes, we revised it.
- As to Figure 5,-7, it will be more interesting to compare FhCaBP4 and FhCaBP2.
- Thank you for your kind comments. FhCaBP2 has only been reported with the C-terminal structure (pdb5FX0). This is why we compare the apo and calcium forms in FhCaBP4. Figure 2 presents a comparison of the C-terminal region between FhCaBP4 and FhCaBP2.
The authors have explored the structure of FhCaBP4, it will be more interesting to test corresponding small molecule compounds on the FhCaBP4, which likely found potential drugs for treatment. è Thank you for your kind comments. Figure 4 represents the initial phase in identifying probable candidates for drug screening.
Conclusion: this part is too long, and should not repeat results, just presented the main fundings and significance of the study in a few sentences.
- Yes, we revised.
“First, to investigate novel parasite-specific techniques, we determined the whole structure of FhCaBP4, a calcium-binding protein derived from F. hepatica. The structure constitutes a homodimer that connects a calmodulin-like domain, including an EF-hand motif at the N-terminus, to a dynein light chain (DLC)-like domain at the C-terminus. Secondly, we propose that EF-hand 2 serves as the primary calcium-binding site, corroborated by biochemical and biophysical investigations, including AlphaFold3 modeling, molecular dynamics simulations, and microscale thermophoresis experiments. Third, when calcium attaches, it stabilizes the structure and exhibits a more distinct unfolding pattern. Conversely, substances such as calmodulin inhibitors or praziquantel demonstrate minimal or insignificant interaction. Collectively, the findings endorse a calcium-gated switching mechanism that conveys EF-hand movements to the DLC-like domain. This switching interface, absent in mammals and seemingly exclusive to trematodes, may provide a valuable foundation for the development of selective inhibitors or immune-targeting methods to address fascioliasis.”
Supplementary files: all the supplementary files should combine in a separate document.
- Yes, we combine in the separate document.

Reviewer 2 Report
Comments and Suggestions for Authors
Dear authors,
This manuscript is well-organized and illustrated, and, in my view, represents highly relevant research.
Major comments:
- The discussion of the results obtained is essentially absent. It is necessary to either introduce a chapter titled 'Discussion' or significantly expand the 'Conclusion' section—please address this.
- In the 'Discussion' chapter, it is also necessary to discuss why it is important to study the calcium-binding proteins of Fasciola hepatica in the context of host-parasite interactions and how these proteins participate in this interaction.
- Also, discuss your findings in the context of other trematodes. Do calcium-binding proteins have a similar role in host-parasite interactions?
- Additionally, please specify the novel contributions of your work, outline the limitations of your study, and discuss in greater depth how the data obtained can be utilized in the future beyond what is mentioned in the 'Conclusion' chapter.
Minor comments:
- Please correct the numerous typos, unnecessary line spacing, and the use of italics where it is not necessary.
- Please provide Table 1 in a higher resolution to enhance its readability."
Author Response
Reviewer2
Dear authors,
This manuscript is well-organized and illustrated, and, in my view, represents highly relevant research.
Major comments:
- The discussion of the results obtained is essentially absent. It is necessary to either introduce a chapter titled 'Discussion' or significantly expand the 'Conclusion' section—please address this.
- In the 'Discussion' chapter, it is also necessary to discuss why it is important to study the calcium-binding proteins of Fasciola hepatica in the context of host-parasite interactions and how these proteins participate in this interaction.
- Also, discuss your findings in the context of other trematodes. Do calcium-binding proteins have a similar role in host-parasite interactions?
- Additionally, please specify the novel contributions of your work, outline the limitations of your study, and discuss in greater depth how the data obtained can be utilized in the future beyond what is mentioned in the 'Conclusion' chapter.
- I appreciate your insightful remarks. In response to other reviewers' comments, we concisely reworked the discussion section and conclusion. Please refer to the bellows.
- Discussion
We determined the first full-length crystal structure of a CaBP4 from F. hepatica (Figure 1, 2). The structure shows a conserved EF-hand calcium-binding motif and a DLC-like domain involved in dimerization bound together in a polypeptide. The full-length structure of FhCaBP2 was examined, although only the C-terminal electron density was detected [10]. It was proposed that FhCaBP2 can be readily cleaved into two domains by protease for cell extraction. Comparable findings were seen for the tegumental-allergen-like proteins (TAL) protein derived from Clonorchis sinensis (CsTALs) [10,12,17]. FhCaBP4 belongs to a trematode-specific family of calcium-binding proteins that function as calcium-sensitive signaling platforms. Comparative studies in other trematodes, such as the TALs in Schistosoma mansoni (SmTALs) and FhCaBPs, have shown similar EF-hand/DLC-like domain combinations, with paralog-specific differences in calcium responsiveness and drug interaction profiles [11-13, 21].
In the calcium binding model of AlphaFold3, EF-hand 2 adopts canonical coordination and superposition with the apo structure shows an outward motion of helices including EF-hand 2 and also increase hydrophobic exposure (Figure 5B). Also, MD analyses show that calcium binding slightly increases Rg and redistributes loop dynamics. These data suggest that EF-hand 2 acts as the primary calcium-binding site and is compatible with a conformational shift brought on by calcium. The flexible β4–β5 loop in FhCaBP4, which differs from its FhCaBP2 paralog, bridge EF-hand helices to the loop of DLC-like domain and an added π–π stack stabilize the loop (Figure 4, 5). Consistently, site-saturation mutagenesis across the 16-residue loop shows broader ΔΔG distributions in FhCaBP4, with the FSHEPF core displaying significantly greater energetic variability than in FhCaBP2, supporting a paralog-specific plasticity module that likely adjusts ligand and partner interactions (Figure 4).
Our ligand-binding assays demonstrate that calcium binds well and clearly affects the protein, but other molecules like the calmodulin antagonists bind weakly at best (Figure 7). Recently, PZQ primarily targets the TRPM ion channel of a parasite, and the ortholog in Fasciola exhibits sequence variations that account for PZQ insensitivity [39, 40]. Figure 7 illustrates that PZQ exhibited minimal binding (Kd > 2000 µM). FhCaBP4 demonstrated the greatest affinity for Ca2+ (Kd = 43.1 ± 21.1 µM), succeeded by CPZ (288.9 ± 99.6 µM), W7 (470.9 ± 157.3 µM), and TFP (699.3 ± 341.2 µM). The observed inflection temperature demonstrates that calcium binding possesses strong affinity and significantly improves protein stability (Table 2). Consequently, these results indicate an alternative function and facilitate the targeting of FhCaBP4 as a component of a parasite-selective approach, utilizing either small-molecule inhibitors or vaccine candidates. Additional comparative analysis of FhCaBP paralogs and other trematode calcium-binding proteins, such as SmTALs or CsTALs, may elucidate if the calcium-mediated alteration serves as a prevalent regulatory mechanism within this family, potentially revealing novel functional or therapeutic insights.
- 5. Conclusions
First, to investigate novel parasite-specific techniques, we determined the whole structure of FhCaBP4, a calcium-binding protein derived from F. hepatica. The structure constitutes a homodimer that connects a calmodulin-like domain, including an EF-hand motif at the N-terminus, to a dynein light chain (DLC)-like domain at the C-terminus. Secondly, we propose that EF-hand 2 serves as the primary calcium-binding site, corroborated by biochemical and biophysical investigations, including AlphaFold3 modeling, molecular dynamics simulations, and microscale thermophoresis experiments. Third, when calcium attaches, it stabilizes the structure and exhibits a more distinct unfolding pattern. Conversely, substances such as calmodulin inhibitors or praziquantel demonstrate minimal or insignificant interaction. Collectively, the findings endorse a calcium-gated switching mechanism that conveys EF-hand movements to the DLC-like domain. This switching interface, absent in mammals and seemingly exclusive to trematodes, may provide a valuable foundation for the development of selective inhibitors or immune-targeting methods to address fascioliasis.
Minor comments:
- Please correct the numerous typos, unnecessary line spacing, and the use of italics where it is not necessary.
- Yes, we checked them and revised.
- Please provide Table 1 in a higher resolution to enhance its readability.
- Yes, we changed the table format as reviewer’s comment.
Round 2
Reviewer 1 Report
Comments and Suggestions for Authors
The authors have addressed most issues in the previous version, and I have no further questions.